# Exploring the Diagnostic and Predictive Value of Oral Microbiome in Esophageal Cancer: A Systematic Review and Meta-Analysis

**DOI:** 10.3390/ijms26199457

**Published:** 2025-09-27

**Authors:** Jie-Chi Chen, Min-Hsun Hsu, Suh-Woan Hu, Yuh-Yih Lin

**Affiliations:** 1School of Dentistry, College of Oral Medicine, Chung Shan Medical University, Taichung 40201, Taiwan; 1130039@live.csmu.edu.tw (J.-C.C.); saekinomao0503@gmail.com (M.-H.H.); 2Institute of Oral Sciences, College of Oral Medicine, Chung Shan Medical University, Taichung 40201, Taiwan; 3Department of Stomatology, Chung Shan Medical University Hospital, Taichung 40201, Taiwan

**Keywords:** oral microbiome, esophageal cancer, association, biomarkers, systematic review, meta-analysis

## Abstract

The research interest in the oral microbiome’s role in esophageal cancer is growing, yet a comprehensive synthesis of available evidence is still lacking. This study aimed to explore the effects of oral microbiome on the development of esophageal cancer through a systematic review of existing literature retrieved from the Embase, PubMed, and Web of Science databases. Eighteen studies published between 2015 and 2024 were obtained, involving 1191 cases and 1403 controls, mostly using oral saliva samples and 16S rRNA gene sequencing. Findings on alpha-diversity were inconsistent, while most studies reported significant beta-diversity differences between cases and controls. Notably, several investigations on esophageal squamous cell carcinoma showed higher levels of *Prevotella*, *Porphyromonas*, and *Fusobacterium*, while two studies on esophageal adenocarcinoma reported elevated levels of *Actinomyces* species. A fixed-effect meta-analysis of two studies showed that individuals with specific oral microbial signatures had significantly higher odds of developing esophageal squamous cell carcinoma (OR = 9.50; 95% CI: 5.89–15.29). Quality assessments highlighted methodological strengths but noted variability in group comparability and local applicability. These results reveal the potential of oral microbiome shift as an early detection biomarker and for developing personalized strategies in treating esophageal cancer, meriting further clinical investigation.

## 1. Introduction

The human microbiota consisting of microbial communities that inhabit various ecological niches throughout the body, including the skin, oral cavity, and gastrointestinal tract, plays an essential role in maintaining overall health [1]. Under normal conditions, these microbial populations coexist in balance with the host. However, when this equilibrium is disturbed, a condition known as dysbiosis, it has been linked to the onset and progression of numerous diseases [1,2]. Although studying the human microbiota has been difficult due to challenges in culturing many microbial species, recent progress in next-generation sequencing technologies and bioinformatics has greatly improved the feasibility of microbiome profiling.

Within the oral cavity, a range of ecological niches are formed by structures such as the teeth, saliva, and various regions of the oral mucosa, including lining, masticatory, and specialized types, which together contribute to the complexity of the oral microbiome [3]. This microbiome comprises more than 700 distinct prokaryotic species, highlighting its ecological diversity and richness [4,5]. A variety of host factors, such as diet habits, alcohol use, smoking, medications, environmental exposures, and genetic background, can influence oral microbiome composition. Disruptions to the oral microbial equilibrium have been associated with several oral diseases, notably dental caries and periodontal diseases [6,7]. Furthermore, growing evidence suggests that oral dysbiosis may be linked to systemic diseases, including respiratory conditions [8], cardiovascular disease [9], rheumatoid arthritis [10], diabetes mellitus [11], and Alzheimer’s disease [12].

Microbial infections have been recognized as important contributors to cancer development, with approximately 15% of all cancer cases attributed to microbes [13]. Among the body’s microbial habitats, the oral cavity acts as a key portal to the gastrointestinal tract, providing a relatively unimpeded path for microorganisms. Consequently, the oral microbiota has garnered increasing attention in the context of gastrointestinal malignancies [14]. An expanding body of evidence indicates a connection between the oral microbiome and cancer, with associations documented in oral squamous cell carcinoma [15], esophageal cancer [16], gastric cancer [17], pancreatic cancer [18], and colorectal cancer [19].

Esophageal cancer is characterized by its high mortality rate and poor prognosis. It ranks as the seventh cause of cancer-related death and is the 11th most diagnosed cancer worldwide [20]. While esophageal squamous cell carcinoma (ESCC) remains the most popular form globally, the incidence of esophageal adenocarcinoma (EAC) is rising rapidly in developed countries [21]. Notably, there is a marked gender disparity in esophageal cancer, with males showing a two- to threefold higher incidence and mortality compared to females [20].

Epithelial tissues of the esophagus and oral cavity are exposed to similar microorganisms through saliva, yet distinct bacterial species may play different roles in the carcinogenesis of oral versus esophageal cancers. For instance, *Streptococcus anginosus* DNA has been detected at significantly higher levels in esophageal cancer tissues compared to oral cancer tissues, suggesting a stronger link between this species and esophageal cancer [22]. Additionally, *Treponema denticola* and *Streptococcus mitis*—two common oral bacteria—have also been found in esophageal cancer samples [22]. These microbes may contribute to esophageal carcinogenesis by promoting inflammatory responses [22,23]. Studies have also shown that bacterial diversity and richness are reduced in the ESCC group compared to controls, and LEfSe analysis has identified *Actinomyces* and *Atopobium* as being potentially associated with increased ESCC risk [24]. In contrast, *Fusobacterium* and *Porphyromonas* were more commonly found in the control group [24]. While these findings indicate a possible microbial link to esophageal cancer, further research is still needed to elucidate the underlying mechanisms.

As the connection between the microbiome and cancer continues to be explored, growing evidence supports the idea that microbial communities may influence both the initiation and progression of various malignancies [14,25]. In particular, several studies have pointed to a possible connection between the oral microbiota and esophageal malignancy [22,23,24,26,27].

Despite these emerging associations, no comprehensive review has systematically evaluated the connection between the oral microbiome and esophageal cancer. The current study conducted a systematic review and meta-analysis of the existing literature. We examined differences in microbial composition and abundance between esophageal cancer cases and healthy individuals, evaluated the strength of these associations, and assessed the potential of oral microbiome as biomarkers for early detection. Ultimately, this study aims to advance our understanding of the role of the oral microbiome in contributing esophageal cancer development and may help to improve more effective therapeutic strategies.

## 2. Methods

### 2.1. Literature Search

This systematic review and meta-analysis were carried out following the Preferred Reporting Items for Systematic Reviews and Meta-Analyses (PRISMA) guidelines. The study protocol was registered in the PROSPERO database (ID1109590). After removing duplicate records, two reviewers (J.-C.C. and M.-H.H.) independently screened the titles and abstracts of the remaining studies to determine eligibility based on predefined inclusion and exclusion criteria. Discrepancies were resolved through discussion with additional reviewers (S.-W.H. and Y.-Y.L.). Full-text articles were subsequently retrieved and evaluated for final inclusion. The study selection process is depicted in the PRISMA 2020 flow diagram (Figure 1), and a completed PRISMA checklist is available in the Appendix A.

A comprehensive literature search was conducted using Embase, PubMed, and Web of Science, encompassing all relevant studies published from each database’s inception up to 6 July 2024.

The search terms for PubMed were as follows:

“(oral[title/abstract] OR saliva[title/abstract] OR tongue[title/abstract] OR salivary[title/abstract]) AND (microbiome[title/abstract] OR microbiota[title/abstract] OR bacteria*[title/abstract] OR microbial[title/abstract] OR microorganism[title/abstract] OR microbe[title/abstract] OR marker[title/abstract] OR organism) AND (Esophageal[title/abstract] OR Esophagus[title/abstract] OR Oesophageal*[title/abstract] OR Oesophagus*[title/abstract]) AND (carcinoma*[title/abstract] OR cancer*[title/abstract] OR neoplas*[title/abstract] OR adenoma*[title/abstract] OR malignan*[title/abstract] OR tumor*[title/abstract] OR tumour*[title/abstract] OR adenocarcinoma*[title/abstract])”.

The search terms for Embase were as follows:

“(oral OR saliva OR tongue OR salivary) AND (microbiome OR microbiota OR bacteria* OR microbial OR microorganism OR microbe OR marker OR organism) AND (esophageal OR esophagus OR oesophagus OR oesophageal) AND (carcinoma* OR cancer* OR neoplas* OR adenoma* OR malignan* OR tumor* OR tumour* OR adenocarcinoma*)”.

The search terms for Web of Science were as follows:

“(TS= ((oral OR saliva OR tongue OR salivary) AND (microbiome OR microbiota OR bacteria* OR microbial OR microorganism OR microbe OR marker OR organism) AND (Esophageal OR Esophagus OR Oesophageal OR Oesophagus) AND (carcinoma* OR cancer* OR neoplas* OR adenoma* OR malignan* OR tumor* OR tumour* OR adenocarcinoma*)))”.

### 2.2. Study Selection

Studies were considered eligible for inclusion based on the following criteria: (1) studies in which subjects are diagnosed with esophageal cancer, irrespective of histopathological subtype (e.g., esophageal adenocarcinoma or esophageal squamous cell carcinoma); (2) case–control studies investigating the association between the oral microbiota and esophageal cancer; (3) interventional studies examining the effect of treatment on the microbiota in esophageal cancer patients. The following exclusion criteria were applied: (1) studies not written in English; (2) publications that did not constitute original research, including literature reviews, meta-analyses, case reports, editorials, conference proceedings, or retracted articles; (3) studies where the subjects are non-human or the research topic does not pertain to esophageal cancer; (4) articles published prior to 31 December 2010; (5) studies for which the full text was unavailable despite reasonable efforts to obtain it.

We restricted the search to articles published from January 2011 to July 2024. Studies prior to 2011 were excluded because most relied on culture-based methods or low-throughput molecular techniques, whereas next-generation sequencing methods (e.g., 16S rRNA sequencing, metagenomics) became widely adopted after 2010, providing higher resolution and comparability for microbiome research. As summarized in Table 1, we defined the PICO elements for our population, intervention/exposure, comparison, and outcomes.

### 2.3. Data Extraction

The following data were extracted from each included study: (1) first author, year of publication, study location (country/region), and study design; (2) cancer type, number of participants, as well as their gender and age; and (3) sample source, targeted 16S rRNA gene region, sequencing platform, and key microbiota-related findings, including alpha-diversity, beta-diversity, and changes in the relative abundance of specific bacterial genera or species. During this process, all relevant tables and figures were carefully examined to ensure comprehensive data collection.

### 2.4. Meta-Analysis

Eligible case–control studies that reported odds ratios (ORs) with 95% confidence intervals (CIs) were identified and included. A quantitative meta-analysis was performed to pool effect estimates (odds ratios with 95% confidence intervals) across eligible studies. Statistical heterogeneity was assessed using Cochran’s Q test (*p* < 0.10 indicating significance) and quantified using the I^2^ statistic, with values of <25% considered low, 25–50% moderate, and > 50% high heterogeneity. In analyses where heterogeneity was negligible (*p* > 0.10 and I^2^ < 25%), a fixed-effect model (Mantel–Haenszel method) was applied, assuming a common effect size. 16 out of 18 studies were excluded from the meta-analysis due to insufficient data. Only two studies provided sufficient quantitative data (effect estimates with confidence intervals or raw case–control abundance data) to be eligible for meta-analysis. A fixed-effect meta-analysis was conducted, and a forest plot was created to illustrate both individual study estimates and the overall pooled effect.

### 2.5. Quality Assessment

The quality of the included studies was assessed using the Critical Appraisal Skills Program (CASP) checklist for case–control studies [28]. This tool consists of 11 questions that evaluate studies across three key domains: (1) the validity of the findings, (2) the clarity and presentation of the results, and (3) the relevance and applicability of the evidence. Each item was rated as “Yes”, “No”, or “Unclear” to help determine the risk of bias and assess the overall reliability of the evidence. Two reviewers (J.-C. Chen and M.-H. Hsu) conducted the quality assessment independently. Disagreements were resolved through discussion, and when necessary, by consultation with a third reviewer (S.-W. Hu) to reach consensus.

## 3. Results

### 3.1. Literature Search and Study Selection

The initial search yielded a total of 2544 articles: 644 from PubMed, 1475 from Embase, and 425 from Web of Science. After removing 551 duplicate records, 1993 unique studies remained for title and abstract screening. The following records were excluded: non-English articles (*n* = 88), review articles (*n* = 393), systematic reviews (*n* = 25), case reports (*n* = 176), editorials (*n* = 1), retracted publications (*n* = 4), non-human studies (*n* = 49), studies unrelated to esophageal cancer (*n* = 126), irrelevant topics (*n* = 644), and studies published before 31 December 2010 (*n* = 432).

A total of 644 were excluded at the title/abstract screening stage for being unrelated to the topic of interest. The most common reasons were that the study examined microbiota from non-oral sites (e.g., gut, gastric), addressed oral microbiota in conditions other than esophageal cancer (e.g., caries, periodontitis), or reported on esophageal cancer without assessing microbial composition (e.g., clinical or genetic studies). Review articles, editorials, and conference abstracts were also excluded at this step.

A total of 55 full-text articles were assessed for eligibility. Of these, additional exclusions were made due to inaccessible full texts (*n* = 11), lack of relevance to esophageal cancer (*n* = 6), absence of oral microbiota data (*n* = 8), missing outcome measures (*n* = 11), and duplicated content (*n* = 1). Ultimately, 18 studies met all inclusion criteria. A detailed summary of the screening process is shown in Figure 1.

### 3.2. Study Characteristics

A total of 18 studies published between 2015 and 2024 were included in this review (Table 2). Most were conducted in China (*n* = 12), followed by the United States (*n* = 4), with single studies from Japan and Taiwan. In terms of cancer type, 11 studies focused on ESCC, two on EAC, one study addressed both subtypes, while two did not specify the histological classification. Additionally, one study examined precancerous esophageal lesions, and another investigated early intramucosal ESCC. 

All included studies employed a case–control design, with two specifically identified as nested case–control studies. The aggregate participant count comprised 1191 individuals in the case groups and 1403 in the control groups. The proportion of males in the case groups was generally higher than in the control groups. Additionally, age differences between cases and controls were commonly observed, with cancer patients typically being older. Sixteen studies reported mean or median age data, showing an age gap of approximately 3 to 15 years favoring older individuals in the case groups. However, a few studies did not report detailed demographic breakdowns.

### 3.3. Sample Collection

The methodological characteristics of the 18 included studies reveal notable variability in sample collection and sequencing strategies (Table 3). All studies employed 16S rRNA gene sequencing to profile the oral microbiota, with the majority utilizing a case–control design.

Saliva was the most collected specimen, appearing in 13 studies, while other sample types included oral wash, tongue coating, buccal mucosa, oral swabs, and oral biofilms. The targeted hypervariable regions of the 16S rRNA gene varied among studies, with V3–V4 being the most frequently amplified (*n* = 9), followed by V4 (*n* = 5), and other regions such as V1–V2 and V3–V5. One study did not report the specific region used. Sequencing platforms also varied across studies, with Illumina-based technologies dominating (e.g., MiSeq, HiSeq 2500, NovaSeq 6000), while earlier or alternative technologies such as 454 Roche FLX and Ion Torrent systems (PGM and S5) were used in earlier publications. This heterogeneity in sequencing methods and 16S regions underscores the need for caution in cross-study comparisons and meta-analytic synthesis, as technical variability may influence microbial profiling outcomes.

### 3.4. Alpha-Diversity and Beta-Diversity

The alpha- and beta-diversity results across the included studies reveal notable heterogeneity (Table 4). Alpha-diversity, most commonly assessed using the Shannon and Chao1 indices, exhibited variable trends. A decrease in diversity among cancer patients was reported in several studies [29,34,42,43], while others noted no significant differences [24,27,32,33,36,39,40,41] or even modest increases [23,30,31,35,37,38], highlighting the complexity of microbial richness and evenness in disease contexts.

In contrast, beta-diversity analyses more consistently revealed significant differences in microbial community composition between case and control groups. Significant beta-diversity was observed in 12 out of 17 studies that reported this metric, employing methods such as UniFrac and Bray–Curtis. Some studies demonstrated statistically significant dissimilarities between groups [24,29,42], while others found no significant differences [27,40]. In some cases, results varied depending on the weighting method applied to the beta-diversity analysis [34].

Overall, while alpha-diversity results remain inconsistent across studies, beta-diversity provides more robust evidence supporting microbial community shifts in esophageal cancer. These findings underscore the utility of community structure-based metrics in distinguishing microbial alterations associated with carcinogenesis.

### 3.5. Microbiome Changes in Esophageal Cancer Patients Compared to Healthy Controls

An overview of microbial composition changes across the included studies reveals notable patterns and recurring taxa associated with esophageal cancer (Table 5). Across both ESCC and EAC, numerous studies reported increased abundances of oral genera such as *Prevotella*, *Porphyromonas*, *Fusobacterium*, and *Streptococcus*, alongside consistent reductions in *Neisseria*, *Lautropia*, and *Corynebacterium*. These shifts were frequently interpreted as potential microbial signatures or biomarkers for ESCC and EAC. Notably, *P. gingivalis* and *F. nucleatum* emerged as recurrently enriched species across multiple datasets, particularly in association with ESCC [30,35]. In addition, some studies highlighted a consistent increase in the ratio of *Prevotella*/*Neisseria* trend in ESCC [32,33].

Beyond case–control comparisons, some studies extended their analysis to precancerous lesions [36] or stratified disease stages [38,42], providing an inside understanding of microbiome dynamics during the cancer development.

Although most studies did not report diagnostic performance metrics, a few reported robust sensitivity and specificity using microbial panels. For example, two studies reported sensitivities ranging from 68.2% to 86.4% and specificities between 58.8% and 96.1%, suggesting that combinations of microbial taxa may offer promising non-invasive screening approaches [35,37]. Although validation in larger, diverse cohorts remains necessary, these findings underscore the translational promise of microbiome-based biomarkers in esophageal cancer detection.

Collectively, these studies support the relevance of specific microbial signatures in esophageal carcinogenesis and highlight emerging microbial biomarkers that may contribute to non-invasive diagnostic tools.

### 3.6. Meta-Analysis of Microbiome-Based Risk Prediction for ESCC

Of the 18 studies included in the systematic review, only two provided sufficient quantitative data (odds ratios with confidence intervals or comparable raw data) to allow pooling in a meta-analysis. The remaining studies were excluded from quantitative synthesis due to incomplete statistical reporting or heterogeneity in outcome measures. As such, the pooled results should be interpreted cautiously given the limited number of eligible studies.

We conducted a fixed-effect meta-analysis of the two independent studies evaluating the association between specific microbial profiles and ESCC risk [35,37]. Individual study ORs ranged from 9.01 to 10.53. The pooled estimate yielded an OR of 9.50 (95% CI: 5.89–15.29), indicating that individuals harboring these microbial signatures have a 9.50-fold higher odds of ESCC compared to controls. This consistent effect across studies supports the robustness of microbial biomarkers in ESCC risk stratification. The corresponding forest plot is presented in Figure 2.

### 3.7. Quality Assessment of the Included Studies

Figure 3 shows the results of the quality assessment of the included studies using the CASP Case–Control Study Checklist. Six studies received more than 9 green lights, while only one study had fewer than 5 green lights. In terms of the checklist items of the CASP Case–Control Study, all studies received a green light for “Did the study address a clearly focused issue?” “Did the authors use an appropriate method to answer their question?” and “Do the results of this study fit with other available evidence?” More than 10 studies received a yellow light for “Aside from the exposure, did the groups have similar characteristics?” “Have the authors taken account of the potential confounding factors in the design and/or in their analysis?” “Was the estimate of the treatment effect precise?” and “Do you believe the results?” Additionally, all studies received a yellow light for “Can the results be applied to your patients/the population of interest?”

## 4. Discussion

To the best of the authors’ knowledge, we present the first systematic review to comprehensively examine the relationship between the oral microbiome and esophageal cancer, with specific focus on both esophageal cancer types, ESCC and EAC. Although findings related to alpha-diversity were inconsistent across studies, several investigations reported significant differences in beta-diversity between patients and healthy controls, indicating alterations in the composition of oral microbial communities. Of particular interest, elevated levels of *Prevotella*, *Porphyromonas*, and *Fusobacterium* were consistently identified in ESCC patients, suggesting that these bacterial taxa may play a role in the development of esophageal cancer.

Several risk factors—such as smoking and diets low in fruits and vegetables—have been linked to both ESCC and EAC. Heavy alcohol consumption has been specifically associated with ESCC, while GERD and Barrett’s esophagus are key risk factors for EAC, although more research is needed to confirm these associations. Obesity appears to elevate EAC risk while being inversely associated with ESCC [44]. On a molecular level, mutations in tumor-related genes including *TP53*, *RB1*, *CDKN2A*, *PIK3CA*, *NOTCH1*, and *NFE2L2* are commonly found in ESCC [45], and abnormal epigenetic modifications are also implicated in its progression [46]. Genomic alterations in ESCC frequently involve the Wnt, cell cycle, Notch, RTK-Ras, and AKT signaling pathways [45].

The microbiome contributes broadly to host health, with the gut microbiome being especially well studied. A balanced gut microbiota supports intestinal barrier function, controls pathogenic bacterial growth, and regulates immune responses to maintain immune homeostasis [47,48,49]. In contrast, microbial imbalance (dysbiosis) can lead to chronic inflammation, metabolic dysregulation, and immune evasion, thereby facilitating cancer development [50,51,52]. Although less explored than its gut counterpart, the oral microbiome is gaining attention for its potential role in the onset and progression of gastrointestinal and systemic cancers, possibly via the oral–gut axis [53,54,55]. This axis presents a plausible mechanism through which oral microbial shifts may influence distant organs, suggesting that the oral microbiome could contribute to tumorigenesis beyond the oral cavity itself.

Some studies have proposed that reduced alpha-diversity in the oral microbiome may be linked to an elevated risk of certain cancers, including lung and breast cancer [56,57,58]. However, this pattern is not consistent across all cancer types. For instance, research on hepatocellular carcinoma has shown increased alpha-diversity in liver and peritumoral tissues, potentially due to the expansion of pathogenic bacteria and resulting microbial evenness [59]. Our systematic review found inconsistent results regarding alpha-diversity between esophageal cancer patients and healthy controls. While some studies reported reduced microbial richness and evenness, others observed no significant differences. These discrepancies may, in part, reflect underlying methodological and biological heterogeneity. From a technical perspective, sample collection methods (e.g., whole saliva versus oral or esophageal swabs) can strongly influence measured diversity, with saliva typically capturing a broader range of planktonic and shed microbial taxa. Similarly, differences in sequencing techniques, including choice of 16S rRNA hypervariable region (V3–V4 vs. V4) and platforms (Illumina MiSeq vs. Ion Torrent), can greatly influence taxonomic precision. Variability in bioinformatic pipelines and diversity indices used across studies further complicates direct comparison. Host-related factors such as location, diet habit, and oral health status also contribute to the observed variability in microbiome. Collectively, these technical and biological factors may account for the heterogeneity observed in alpha-diversity outcomes. To improve reproducibility and comparability, future research would benefit from adopting standardized protocols for sample collection, sequencing, and analysis.

Prior studies on chronic pancreatitis have revealed marked beta-diversity differences in gut microbiota compared to healthy individuals, consistent across varying disease causes [60,61]. Moreover, reductions in beneficial bacteria were found to contribute to microbiome imbalances in chronic pancreatitis [62]. Similarly, several studies included in this review identified notable beta-diversity differences in the oral microbiomes of ESCC or EAC patients versus controls, suggesting that microbial composition changes may reflect the disease process rather than directly cause it. Further research is needed to pinpoint the specific microbial patterns associated with ESCC and EAC to better understand their roles in cancer progression.

Beyond taxonomic changes, specific microbial components and metabolites may directly influence esophageal carcinogenesis. For instance, *P. gingivalis* LPS activates TLR4–NF-κB signaling, sustaining inflammation, while *F. nucleatum* FadA adhesin binds E-cadherin, activating β-catenin pathways and promoting proliferation and immune evasion. Metabolites such as butyrate (effects on host cell proliferation) and hydrogen sulfide (genotoxic and pro-oncogenic effects) further implicate microbial metabolism in ESCC. In contrast, mechanistic data for EAC remain limited, with only preliminary links to nitrate-reducing taxa. Integrating taxonomic, metagenomic, and metabolomic findings will be critical for clarifying functional pathways and host–microbe interactions. 

The oral microbiome may influence carcinogenesis through multiple mechanisms. First, certain microbes can induce chronic inflammation, promoting abnormal cell growth, gene mutations, and activation of oncogenes. Second, bacterial interference with host regulatory pathways—such as enhanced proliferation, cytoskeletal remodeling, NF-κB signaling activation, and inhibition of apoptosis—may contribute to tumor progression. Third, some bacteria produce carcinogenic metabolites that directly damage host cells [63]. Elevated levels of *Prevotella* have been linked to both local and systemic diseases. This genus can activate TLR2 on dendritic cells, leading to secretion of cytokines such as IL-23 and IL-1β, which polarize Th17 cells. This, in turn, stimulates epithelial cells to release IL-6, IL-8, and CCL20, driving neutrophil recruitment and local inflammation [64,65]. In our analysis, *Prevotella* levels were consistently elevated in the oral microbiome of ESCC patients, supporting its possible role in promoting chronic inflammation and contributing to ESCC risk.

Our review also identified a notable increase in *Porphyromonas* species, particularly *P. gingivalis*, within the oral microbiota of patients with ESCC. *P. gingivalis*, a Gram-negative anaerobic bacterium commonly linked to periodontal disease, has also been associated with various gastrointestinal cancers [66,67,68]. Its potential role in tumor development is thought to involve activation of TLR, NF-κB, and MAPK signaling pathways, as well as suppression of apoptosis [63,69]. Mechanistically, *P. gingivalis* upregulates the anti-apoptotic gene *Bcl-2*, inhibits pro-apoptotic proteins such as Bad and caspase-9, and activates the Jak1/Akt/Stat3 pathway, thereby interfering with intrinsic mitochondrial apoptosis [63,70]. However, since *P. gingivalis* is a common resident of the oral microbiota, its involvement in carcinogenesis likely depends on its interaction with other microbial species or host-related factors [71]. Further research is needed to clarify these interactions and the specific mechanisms by which *P. gingivalis* may contribute to the development of esophageal cancer.

In addition, accumulating evidence suggests that *F. nucleatum* may also play a role in tumorigenesis. This organism produces outer membrane vesicles that activate TLR4 and downstream signaling cascades, including p-ERK, p-CREB, and NF-κB, which stimulate epithelial cells to secrete pro-inflammatory cytokines. Such inflammatory activity has been implicated in the development of oral squamous cell carcinoma and colorectal cancer [72,73]. In the present study, elevated levels of *F. nucleatum* were also observed in the oral microbiota of ESCC patients, suggesting that it may migrate to the esophagus, where it could promote local inflammation and potentially contribute to cancer progression.

While our review primarily focused on taxonomic changes, it is increasingly clear that functional shift in the microbiome may be equally or more relevant to esophageal carcinogenesis. Recent metagenomic and metabolomic studies suggest that dysbiosis in the oral microbiome is accompanied by shifts in metabolic potential, including enrichment of pathways related to lipopolysaccharide biosynthesis, nitrate/nitrite reduction, and sulfur metabolism. These functional changes can contribute to chronic inflammation and DNA damage within the esophageal epithelium. Moreover, several reports have highlighted altered production of short-chain fatty acids such as butyrate, which, while generally anti-inflammatory, may exert context-dependent effects on epithelial proliferation and tumor progression. Similarly, hydrogen sulfide, produced by oral anaerobes, has been implicated as a genotoxic and pro-oncogenic metabolite. Integrating such functional insights with taxonomic profiles provides a more mechanistic perspective, linking specific microbial communities to host signaling pathways, immune modulation, and tumor biology. Future research should therefore incorporate multi-omics approaches (e.g., shotgun metagenomics, metatranscriptomics, metabolomics) to capture both arms of microbial composition and functional capacity, for a more comprehensive understanding of how microbial dysbiosis contributes to esophageal cancer.

The CASP quality assessment indicated that most of the included studies were of moderate to high methodological quality, with clearly defined objectives and appropriate designs. However, variability was noted in several domains, particularly group comparability and control for confounders. While many studies accounted for age and gender, fewer consistently adjusted for smoking and alcohol consumption, despite their strong associations with both esophageal cancer risk and oral microbiome composition. In some cases, incomplete reporting of these variables limited the ability to assess their impact. This heterogeneity in adjustment for key confounders represents a significant limitation of the current evidence base, as residual confounding may partly explain the observed microbial associations. Future studies should therefore adopt more rigorous designs with comprehensive recording and adjustment for demographic and lifestyle factors in order to strengthen causal inference.

Our initial search covered studies published up to mid-2024. The databases were re-checked in July 2025 prior to submission; 4 additional studies were identified with consistent and additional informative findings [74,75,76,77]. The oral microbiome also shows promise as a non-invasive biomarker for EC. Predictive models based on microbial profiles have achieved high accuracy, with an ESCC classifier reaching an AUC of 0.87. Key microbial markers include *Neisseria perflava* and *Haemophilus parainfluenzae*. Saliva-based prediction models also demonstrated strong discriminative ability (AUC = 0.791), with decreased abundance of *Akkermansia* and *Escherichia-Shigella* linked to increased cancer risk. These findings underscore the translational potential of microbial markers for early detection.

An important limitation of the current evidence base is the scarcity and inconsistency of studies focused on EAC. While the majority of available research has investigated ESCC, only a few studies have examined EAC, and their results are mixed, with some suggesting enrichment of *Actinomyces* species or nitrate-reducing taxa while others reported no significant differences. Given the distinct etiologies, histopathology, and risk factors of ESCC and EAC, findings from ESCC cannot be readily applied to all esophageal cancer types. Overinterpretation of ESCC-associated taxa as universal biomarkers may therefore be misleading. Future studies should prioritize EAC-specific investigations, ideally with larger cohorts and standardized methodologies, to clarify whether distinct microbial signatures or functional pathways contribute to its pathogenesis.

While many studies have linked oral microbiome dysbiosis to esophageal cancer, a causal relationship has not been clearly established. It is possible that the relationship is bidirectional [78]. While some microbes may contribute to cancer development, the tumor environment itself could favor the growth of specific microbial communities. For example, *Streptococcus bovis* or its antigens have been shown to accelerate precancerous lesion progression in the colon but not in healthy rats, suggesting a role only in already altered tissues [79]. In esophageal cancer, tumor progression may lead to metabolic and inflammatory changes that reshape the microbiota—favoring opportunistic pathogens and suppressing beneficial species. More studies are needed to explore these temporal and mechanistic dynamics. Although our meta-analysis demonstrates a strong association between oral microbiota alterations and esophageal cancer, causality cannot be inferred from observational data alone. Tumor-associated changes (such as hypoxia, altered immune signaling, and disrupted tissue architecture) could create ecological niches that favor colonization by anaerobic and opportunistic taxa like *Fusobacterium* or *Porphyromonas*. These observations suggest that some microbial signatures may be consequences rather than causes of tumor development.

However, emerging evidence from causal inference approaches provides additional clarity. A recent Mendelian randomization (MR) study [80] identified significant causal relationships between specific oral microbial taxa and esophageal cancer, while reverse MR analyses did not support a causal effect of esophageal cancer on the oral microbiome. Taken together, current evidence suggests a bidirectional interplay, with microbial dysbiosis potentially contributing to esophageal carcinogenesis while established tumors may further modify local microbial ecology.

A key limitation of the current study is the insufficient control for major confounding factors that are determinants of both oral microbiome composition and esophageal cancer risk. For example, smoking and alcohol consumption can directly alter microbial communities by increasing oxidative stress, impairing mucosal immunity, and selecting for anaerobic taxa, while also acting as independent carcinogens. Similarly, age influences oral microbial diversity and immune function, potentially shaping observed case–control differences. The widespread use of proton pump inhibitors in patients with gastroesophageal reflux may also have oral and esophageal microbiota shift by altering gastric acidity and bacterial colonization. Although many included studies mentioned these variables, few adjusted for them comprehensively in their analyses, and others reported limited information on participant lifestyle or medication history. This lack of adjustment complicates causal interpretation, raising the possibility that some of the associations attributed to microbial taxa may instead reflect residual confounding. To strengthen the evidence, future research should incorporate detailed assessment of lifestyle and medication exposures, apply multivariable models to adjust for confounders, and, ideally, adopt longitudinal designs to disentangle whether microbial changes precede or follow esophageal carcinogenesis. The studies reviewed here showed considerable heterogeneity, driven by differences in lifestyle, geography, and genetic background across populations. This variability limits the ability to generalize findings and draw uniform conclusions. Despite these limitations, our review provides early evidence of a potential link between oral microbiota and esophageal cancer. However, due to small sample sizes and methodological differences across studies, future large-scale, high-quality research is essential to confirm these associations and strengthen the reliability of the findings.

## 5. Conclusions

This study systematically reviewed the current evidence on the connection between the oral microbiome and esophageal cancer, highlighting specific bacterial taxa, including *Prevotella*, *Porphyromonas*, and *Fusobacterium*, with potential utility as diagnostic biomarkers. Oral microbiome-based biomarkers hold promise for early screening and risk assessment, thereby enhancing the early diagnostic capabilities of esophageal cancer. Moreover, microbiome-targeted therapeutic strategies may become part of the realm of precision medicine. To advance the field, future research should adopt standardized protocols for sample collection, sequencing, and data analysis to minimize technical heterogeneity and improve comparability across cohorts. Longitudinal designs are essential to clarify the temporal sequence of dysbiosis and cancer onset, helping to distinguish microbial drivers from passengers. Incorporating integrated multi-omics approaches (e.g., shotgun metagenomics, metatranscriptomics, metabolomics, and host transcriptomics) will be critical for linking microbial taxa to functional pathways and host responses, thereby illuminating mechanisms of carcinogenesis. Paying more attention to confounding factors such as smoking, alcohol use, age, and medication history is also necessary to strengthen causal inference. Finally, expanding research to include more EAC-focused studies will be vital to determine whether distinct microbial signatures contribute to its pathogenesis. Additionally, future research should focus on elucidating the interactions between the microbiome and esophageal cancer development, thereby enhancing the potential clinical applications of microbiome-based approaches in the prevention and treatment of esophageal cancer.

## Figures and Tables

**Figure 1 ijms-26-09457-f001:**
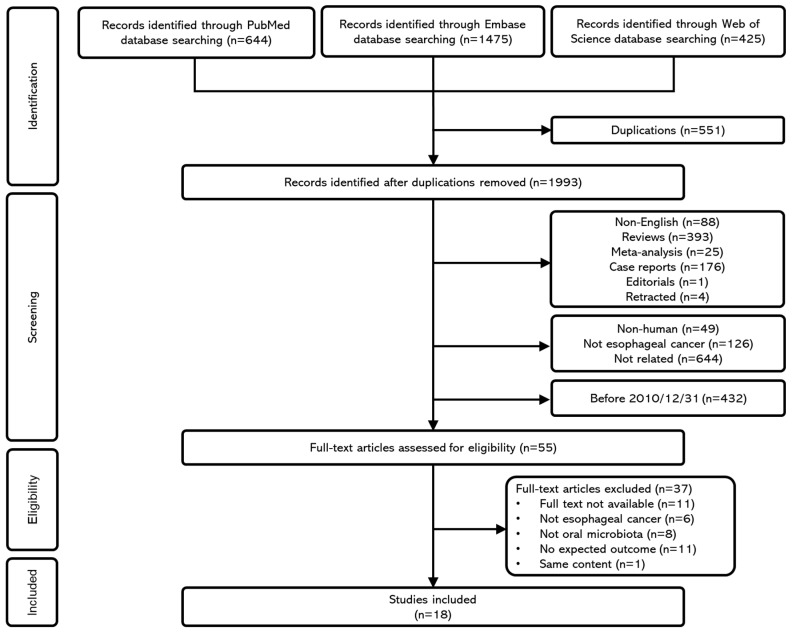
PRISMA flow diagram illustrating the process of study identification, screening, inclusion, and exclusion.

**Figure 2 ijms-26-09457-f002:**
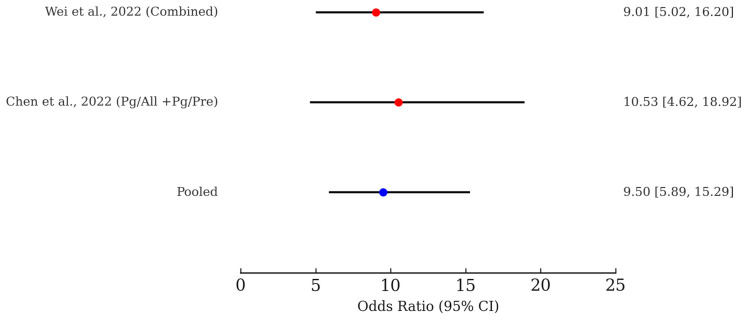
Forest plot depicting ORs and 95% CIs for microbiome-based risk prediction of ESCC. Individual study estimates from Wei et al. (2022) [35] and Chen et al. (2022) [37] are shown in red. A fixed-effect meta-analysis generated a pooled OR of 9.50 (95% CI: 5.89–15.29) is shown in blue.

**Figure 3 ijms-26-09457-f003:**
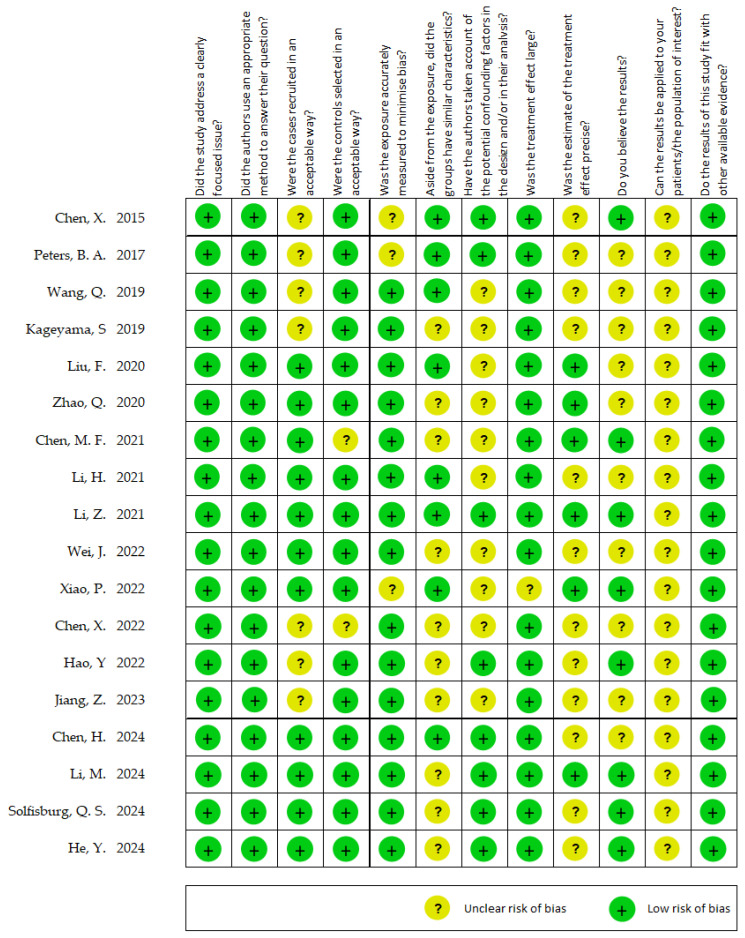
Quality assessment using the CASP Case–Control Study Checklist [23,24,27,29,30,31,32,33,34,35,36,37,38,39,40,41,42,43].

**Table 1 ijms-26-09457-t001:** PICO framework for oral microbiota analysis in esophageal cancer.

Component	Description
P (Population)	Patients diagnosed with esophageal cancer (including ESCC and EAC) as well as healthy controls.
I (Intervention/Exposure)	Analysis of oral microbiota composition, primarily via 16S rRNA gene sequencing from saliva or other oral samples (e.g., oral swabs, tongue coating).
C (Comparison)	Comparison of oral microbiome profiles between esophageal cancer patients and healthy controls.
O (Outcome)	Differences in microbial diversity (alpha- and beta- diversity), abundance of specific taxa, and diagnostic performance metrics such as sensitivity, specificity, and odds ratios for microbial biomarkers.

ESCC: esophageal squamous cell carcinoma, EAC: esophageal adenocarcinoma.

**Table 2 ijms-26-09457-t002:** Demographics of included studies.

First Author	Year	Country	Study Design	Cancer Type	Participants (Case/Control)	Sex (Male %) (Case/Control)	Age (Case/Control)	Reference
Chen, X.	2015	China	Case–control study	ESCC	87/85	67.82/72.94	64.8/66 ^a^	[29]
Peters, B.A.	2017	USA	Nested case–control study	EAC and ESCC	EAC: 81/160, ESCC: 25/50	EAC: 92.6/92.5, ESCC: 40/40	EAC: 68.0/62.4 ^a^ ESCC: 66.6/66.8 ^a^	[27]
Wang, Q.	2019	China	Case–control study	ESCC	20/21	70.0/57.14	65.9/65.14 ^a^	[24]
Kageyama, S.	2019	Japan	Case–control study	Esophageal cancer	12/118	66.7/71.2	68.4/66.4 ^a^	[30]
Liu, F.	2020	China	Nested case–control study	ESCC	84/168	52.38/52.38	57/56 ^b^	[31]
Zhao, Q.	2020	China	Case–control study	Esophageal cancer	39/51	59.0/45.1	60.39/49.18 ^a^	[32]
Chen, M. F.	2021	Taiwan	Case–control study	ESCC	34/18	//	//	[23]
Li, H.	2021	China	Case–control study	ESCC	33/35	84.9/65.7	66/61 ^b^	[33]
Li, Z.	2021	China	Case–control study	ESCC	70/82	65.7/58.5	63.64/58.51 ^a^	[34]
Wei, J.	2022	China	Case–control study	ESCC	178/101	78.09/49.50	Screening: 61.71/43.90 ^a^, Verification: 61.41/44.45 ^a^	[35]
Xiao, P.	2022	China	Case–control study	Esophageal Precancerous Lesions	123/176	57.7/54.5	59.71/58.99 ^a^	[36]
Chen, X.	2022	China	Case–control study	ESCC	90/50	78.89/54.00	60.8/47.7 ^a^	[37]
Hao, Y.	2022	USA	Case–control study	EAC	19/27	94.74/62.96	59.9/56.3 ^a^	[38]
Jiang, Z.	2023	China	Case–control study	ESCC	56/53	53.57/49.06	54.56/49.32 ^a^	[39]
Chen, H.	2024	China	Case–control study	Early-stage intramucosal ESCC	31/21	61.3/52.4	70/64 ^b^	[40]
Li, M.	2024	China	Case–control study	ESCC	52/52	//	//	[41]
Solfisburg, Q.S.	2024	USA	Case–control study	EAC	78/125	79.49/36.00	65/50 ^b^	[42]
He, Y.	2024	China	Case–control study	ESCC	Before: 79/10, After: 8/10	Before: 74.6/50.0, After: 50.0/50.0	Before: 72/70 ^b^ After: 75/70 ^b^	[43]

ESCC: esophageal squamous cell carcinoma, EAC: esophageal adenocarcinoma, ^a^ Mean age. ^b^ Median age.

**Table 3 ijms-26-09457-t003:** Summary of included study methods.

First Author	Year	Sample Type	16S Region	Sequencing Platform	Reference
Chen, X.	2015	Saliva	V3–V4	454 Roche FLX Titanium adapters (454 Life Sciences, Branford, CT, USA)	[29]
Peters, B.A.	2017	Oral wash	V4	Illumina MiSeq (-)	[27]
Wang, Q.	2019	Saliva	V3–V4	Illumina MiSeq (Illumina, San Diago, CA, USA)	[24]
Kageyama, S.	2019	Saliva	V1–V2	Ion PGM Hi-Q view (Thermo Fisher Scientific, Waltham, MA, USA)	[30]
Liu, F.	2020	Oral swabs	V3–V4	Ion S5 XL (Thermo Fisher Scientific, Waltham, MA, USA)	[31]
Zhao, Q.	2020	Saliva	V3–V4	Illumina MiSeq PE250 (Illumina, San Diago, CA, USA)	[32]
Chen, M. F.	2021	Oral biofilms	Not reported	Illumina MiSeq (Illumina, San Diago, CA, USA)	[23]
Li, H.	2021	Saliva	V3–V4	Illumina MiSeq 2×300 bp (Illumina, San Diago, CA, USA)	[33]
Li, Z.	2021	Saliva	V4	Ion S5TM XL (Thermo Fisher Scientific, MA, USA)	[34]
Wei, J.	2022	Saliva	V4	Illumina HiSeq 2500 (Illumina, San Diago, CA, USA)	[35]
Xiao, P.	2022	Tongue coating	V3–V4	Illumina MiSeq PE (Illumina, San Diago, CA, USA)	[36]
Chen, X.	2022	Saliva	V4	Illumina HiSeq2500 (Illumina, San Diago, CA, USA)	[37]
Hao, Y.	2022	Buccal mucosa	V3–V5	Not reported	[38]
Jiang, Z.	2023	Oral swabs	V3–V4	Illumina NovaSeq6000 PE250 (Illumina, San Diago, CA, USA)	[39]
Chen, H.	2024	Saliva	V3–V4	Illumina NovaSeq (Illumina, San Diago, CA, USA)	[40]
Li, M.	2024	Saliva	V4	Illumina MiniSeq (Illumina, San Diago, CA, USA)	[41]
Solfisburg, Q.S.	2024	Saliva	V3–V4	Illumina MiSeq (Illumina, San Diago, CA, USA)	[42]
He, Y.	2024	Saliva	V3–V4	Illumina NovaSeq 6000 (Illumina, San Diago, CA, USA)	[43]

**Table 4 ijms-26-09457-t004:** Alpha- and beta-diversity of included studies.

First Author	Alpha Diversity	Beta Diversity	Reference
Chen, X.	↓ Shannon (3.7 → 3.4), Chao1 (147.2 → 120.8)	Significant (*p* < 0.05, UniFrac)	[29]
Peters, B.A.	Not significant	Not significant (UniFrac)	[27]
Wang, Q.	Not significant	Significant (*p* = 0.037, Bray-Curtis)	[24]
Kageyama, S.	↑ Shannon (~3.4 → ~3.6), Chao1 (~200 → ~220)	Significant (*p* = 0.01, UniFrac)	[30]
Liu, F.	↑ Shannon	Not significant (UniFrac)	[31]
Zhao, Q.	Not significant	Significant (*p* = 0.001, Bray-Curtis)	[32]
Chen, M. F.	↑ Shannon	Significant (*p* = 0.001)	[23]
Li, H.	Not significant	Unweighted: significant (*p* = 0.001), Weighted: not significant	[33]
Li, Z.	↓ Shannon (~6.1 → ~5.8)	Significant (*p* = 0.001, Bray-Curtis)	[34]
Wei, J.	↑ *S. salivarius*, etc. (no Shannon shown)	Not reported	[35]
Xiao, P.	Not significant	Not reported	[36]
Chen, X.	↑ Shannon	Not significant	[37]
Hao, Y.	↑ Shannon	Significant (*p* < 0.01, UniFrac)	[38]
Jiang, Z.	Not significant	Significant (*p* < 0.05)	[39]
Chen, H.	Not significant	Not significant (Bray-Curtis)	[40]
Li, M.	Not significant	Significant (*p* < 0.01, UniFrac)	[41]
Solfisburg, Q.S.	↓ Shannon, ↓ Simpson	Significant (*p* < 0.01, UniFrac)	[42]
He, Y.	↓ Chao1	Significant (Bray-Curtis)	[43]

Arrows indicate the direction of change in diversity indices when comparing esophageal cancer patients with healthy controls (↑ = increased; ↓ = decreased).

**Table 5 ijms-26-09457-t005:** Microbiota comparison of included studies.

First Author	Key Microbial Changes		Potential Biomarkers	Sensitivity/Specificity	Notes	Reference
	Genus	Species				
Chen, X.	*Prevotella* (↑), *Streptococcus* (↑), *Porphyromonas* (↑); *Lautropia* (↓), *Corynebacterium* (↓)	*-*	*Prevotella*, *Streptococcus*, *Porphyromonas*	Not reported	First ESCC saliva study; 454 sequencing	[29]
Peters, B.A.	-	*Porphyromonas gingivalis* (↑), *Prevotella nanceiensis* (↑), *Treponema vincentii* (↑)	*Porphyromonas gingivalis*, *Prevotella nanceiensis*	Not reported	ESCC and EAC separation	[27]
Wang, Q.	*Actinomyces* (↑), *Atopobium* (↑); *Fusobacterium* (↓), *Porphyromonas* (↓)	*-*	*Actinomyces*, *Atopobium*	Not reported	Small sample; diversity not significant	[24]
Kageyama, S.	-	*Porphyromonas gingivalis* (↑), *Fusobacterium nucleatum* subsp. *vincentii* (↑)	*Porphyromonas gingivalis*, *Fusobacterium nucleatum*	Not reported	OTU-rich profile in EC	[30]
Liu, F.	-	*Fusobacterium nucleatum* (↑), *Actinomyces naeslundii* (↑), *Prevotella intermedia* (↑), *Treponema vincentii* (↑)	*Fusobacterium nucleatum*, *Actinomyces naeslundii*	Not reported	Enriched taxa profiled	[31]
Zhao, Q.	*Prevotella* (↑); *Neisseria* (↓)	*-*	*Prevotella*, *Neisseria*	Not reported	Clear Prevotella ↑/Neisseria ↓ pattern	[32]
Chen, M. F.	-	*Porphyromonas gingivalis* (↑), *Veillonella parvula* (↑)	*Porphyromonas gingivalis*, *Veillonella parvula*	Not reported	Clinical + mechanistic focus	[23]
Li, H.	*Streptococcus* (↑), *Prevotella_7* (↑); *Neisseria* (↓)	*-*	*Streptococcus*, *Prevotella_7* *	Not reported	Diversity mixed; taxa significant	[33]
Li, Z.	*Parvimonas* (↑), *Helicobacter* (↑), *Peptostreptococcus* (↑)	*-*	*Parvimonas*, *Helicobacter*	Not reported	Disease progression correlated	[34]
Wei, J.	-	*Streptococcus salivarius* (↑), *Fusobacterium nucleatum* (↑), *Porphyromonas gingivalis* (↑)	*Streptococcus salivarius*, *Fusobacterium nucleatum*, *Porphyromonas gingivalis*	69.3–86.4%/58.8–96.1%	qPCR validation used	[35]
Xiao, P.	*Capnocytophaga* (↑); *Atopobium* (↓), *Hydrobacter* (↓)	*Eubacterium yurii* (↑)	*Eubacterium yurii*, *Capnocytophaga*	Not reported	Focused on pre-cancerous lesion comparison	[36]
Chen, X.	*Leptotrichia* (↑), *Porphyromonas* (↑)	*-*	*Leptotrichia*, *Porphyromonas*	68.2–86.4%/64.4–86.0%	V4 analysis; genus-level resolution	[37]
Hao, Y.	-	*Actinomyces bowdenii* (↑), *Atopobium parvulum* (↑)	*Actinomyces bowdenii*, *Atopobium parvulum*	Not reported	Staged analysis with species detail	[38]
Jiang, Z.	*Leptotrichia* (↑)	*-*	*Leptotrichia*	Not reported	Genus-focused significant taxa	[39]
Chen, H.	*Shigella* (↑), *Leptotrichia* (↑)	*Porphyromonas endodontalis* (↑)	*Porphyromonas endodontalis*, *Leptotrichia*	Not reported	Species-level marker clarity	[40]
Li, M.	-	*Prevotella histicola* (↑), *Fusobacterium nucleatum* (↑), *Prevotella intermedia* (↑)	*Prevotella* spp., *Fusobacterium nucleatum*	Not reported	Expanded Prevotella panel	[41]
Solfisburg, Q.S.	*Streptococcus* spp. (↑)	-	*Streptococcus* spp.	Not reported	Dysplasia stratification	[42]
He, Y.	*Veillonellaceae* (↑)	*Prevotella salivae* (↑)	*Prevotella salivae*, *Veillonellaceae*	Not reported	Pre-/post-treatment comparison	[43]

Arrows indicate the direction of change in diversity indices when comparing esophageal cancer patients with healthy controls (↑ = increased; ↓ = decreased). ESCC: esophageal squamous cell carcinoma, EAC: esophageal adenocarcinoma, EC: esophageal cancer. OTU: operational taxonomic unit. * *Prevotella_7* is a sub-genus level taxonomic unit in 16S rRNA gene sequencing outputs within the broader *Prevotella* genus.

## Data Availability

Data are available upon reasonable request.

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
