# Peer review of "Exploring the Diagnostic and Predictive Value of Oral Microbiome in Esophageal Cancer: A Systematic Review and Meta-Analysis"

_ijms, 2025, doi:10.3390/ijms26199457_

Round 1
Reviewer 1 Report
Comments and Suggestions for Authors
This systematic review and meta-analysis provides a timely and comprehensive synthesis of the emerging evidence linking the oral microbiome to esophageal cancer (ESCC and EAC). The study is well-structured, follows PRISMA guidelines, and offers valuable insights into microbial diversity and taxonomic shifts associated with disease. However, the review would be significantly strengthened by a more detailed analysis of the mechanistic pathways through which specific microbes or their metabolites contribute to carcinogenesis, as well as clearer reporting of taxonomic levels in key results tables.
(1) The taxonomic level (genus, species, etc.) for the "Key Microbial Changes" and "Potential Biomarkers" listed in Table 5 is not consistently clear or explicitly stated, which hinders the interpretation and comparison of results across studies. This should be standardized and specified for each entry.
(2) The discussion section would be significantly strengthened by including a dedicated synthesis of current mechanistic evidence, summarizing how specific microbial components (e.g., LPS of P. gingivalis, FadA adhesin of F. nucleatum) or metabolites (e.g., butyrate, hydrogen sulfide) are proposed to influence ESCC/EAC pathogenesis through pathways like chronic inflammation, immune evasion, or activation of oncogenic signaling.
(3) The high heterogeneity in alpha-diversity findings is noted but not sufficiently explored; a brief discussion on potential technical or biological reasons for these inconsistencies (e.g., sample collection methods, sequencing platforms, bioinformatic pipelines) would provide valuable context.
(4) The review focuses on taxonomic changes but omits discussion of functional shifts in the microbiome; incorporating findings from metagenomic or metabolomic studies where available would greatly enhance the understanding of the mechanistic role of the oral microbiome in esophageal carcinogenesis.
(5) The limited number and inconsistent results of studies focused on EAC are a significant limitation that should be more explicitly highlighted in the discussion to caution against overgeneralizing findings from ESCC to all esophageal cancer types.
(6) The potential impact of major confounding factors (e.g., smoking, alcohol, age, PPI use) on the observed microbiome differences is acknowledged in the quality assessment but should be more deeply discussed as a key limitation affecting the current evidence base and the interpretation of causality.
(7) While the meta-analysis shows a strong association, the discussion on causality is brief; a more nuanced exploration of the potential for reverse causality (i.e., the tumor microenvironment shaping the microbiome) would provide a more comprehensive perspective.
(8) The conclusion would benefit from more specific and actionable recommendations for future research, such as the need for standardized protocols, longitudinal designs to establish temporality, and integrated multi-omics approaches to link microbial taxa to function and mechanism.
Author Response
Comment 1:
The taxonomic level (genus, species, etc.) for the "Key Microbial Changes" and "Potential Biomarkers" listed in Table 5 is not consistently clear or explicitly stated, which hinders the interpretation and comparison of results across studies. This should be standardized and specified for each entry.
Response 1:
We thank the reviewer for this observation. We have revised Table 5 accordingly to ensure that the taxonomic level is explicitly indicated for every microbial entry. This standardization allows for clearer interpretation and comparability across studies.
Comment 2:
The discussion section would be significantly strengthened by including a dedicated synthesis of current mechanistic evidence, summarizing how specific microbial components (e.g., LPS of P. gingivalis, FadA adhesin of F. nucleatum) or metabolites (e.g., butyrate, hydrogen sulfide) are proposed to influence ESCC/EAC pathogenesis through pathways like chronic inflammation, immune evasion, or activation of oncogenic signaling.
Response 2:
We thank the reviewer for this suggestion. We have incorporated this point in the Discussion (Page15, Line402) about Mechanistic links between oral microbiota and esophageal carcinogenesis.
Comment 3:
The high heterogeneity in alpha-diversity findings is noted but not sufficiently explored; a brief discussion on potential technical or biological reasons for these inconsistencies (e.g., sample collection methods, sequencing platforms, bioinformatic pipelines) would provide valuable context.
Response 3:
We thank the reviewer for the suggestion. We have thus rewrite one of the paragraphs in the Discussion (Page14, Line376) to address the high heterogeneity in alpha diversity.
Comment 4:
The review focuses on taxonomic changes but omits discussion of functional shifts in the microbiome; incorporating findings from metagenomic or metabolomic studies where available would greatly enhance the understanding of the mechanistic role of the oral microbiome in esophageal carcinogenesis.
Response 4:
We thank the reviewer for the valuable suggestion. We have thus added a new paragraph in the Discussion (Page15, Line445) to discuss functional shifts in the microbiome.
Comment 5:
The limited number and inconsistent results of studies focused on EAC are a significant limitation that should be more explicitly highlighted in the discussion to caution against overgeneralizing findings from ESCC to all esophageal cancer types.
Response 5:
We thank the reviewer for this observation. We have revised the paragraph in the Discussion (Page16, Line473) to discuss EAC studies.
Comment 6:
The potential impact of major confounding factors (e.g., smoking, alcohol, age, PPI use) on the observed microbiome differences is acknowledged in the quality assessment but should be more deeply discussed as a key limitation affecting the current evidence base and the interpretation of causality.
Response 6:
We thank the reviewer for the suggestion. We added a paragraph in Discussion (Page 17, Line507) to discuss these confounding factors.
Comment 7:
While the meta-analysis shows a strong association, the discussion on causality is brief; a more nuanced exploration of the potential for reverse causality (i.e., the tumor microenvironment shaping the microbiome) would provide a more comprehensive perspective.
Response 7:
We thank the reviewer for the suggestion. We added a paragraph in Discussion (Page 16, Line492) to discuss Causality and Reverse Causality
Comment 8:
The conclusion would benefit from more specific and actionable recommendations for future research, such as the need for standardized protocols, longitudinal designs to establish temporality, and integrated multi-omics approaches to link microbial taxa to function and mechanism.
Response 8:
We thank the reviewer for the suggestion. We have thus revised our conclusion (Page17)
Reviewer 2 Report
Comments and Suggestions for Authors
I would like to thank the Editors for the opportunity to review this manuscript.
This article presents a systematic review and meta-analysis investigating the relationship between the oral microbiome and oesophageal cancer. It synthesises the results of 18 studies, highlighting patterns of microbial diversity and key bacterial taxa such as Prevotella, Porphyromonas and Fusobacterium, and discusses their potential as diagnostic biomarkers and contributors to carcinogenesis.
- Materials and methods: The end date of the search was set at 6 July 2024, i.e., mid-2024. Please provide information on whether the search was updated close to the date of manuscript submission. Such information would increase the timeliness of the study.
- Materials and methods: It would be useful to supplement this section with more detailed information on the meta-analysis methods. The information provided is not detailed enough. Please explain why the fixed-effect model was chosen? Was heterogeneity assessed using specific tests?
- Materials and methods: Please indicate at this stage how many studies were included in the meta-analysis and whether any were excluded due to insufficient data (in the next section there are 18).
- Materials and methods: The CASP checklist was used, which is an appropriate choice. However, it would be worth adding information on whether the CASP assessments were carried out independently by two reviewers.
- Materials and methods: Please indicate why articles published before 31 December 2010 were excluded (is it about NGS?).
- Results: Please provide a definition of the exclusion step ‘irrelevant topics’ in the results, in the table - Not related (n = 644); this aspect is unclear. Please consider specifying common topics or categories of irrelevance.
- Results: Table 2 Abbreviations (e.g., ESCC, EAC) should be defined once in a footnote to the table, even if they have been explained earlier in the text.
- Results: Table 4. Please try to standardise the entries in the table and remove the inconsistent formatting. Some rows contain detailed statistics (e.g., Shannon values), while others contain only a qualitative description, “No α diff”. Standardisation is recommended (e.g., mean ± SD or at least consistent directional reporting). Abbreviations such as “diff.” or “NS” should be replaced with full terms (“difference”, “not significant”).
- Results: Table 4. The description for Table 4 mentions ‘significant’ results but does not provide consistent p-value thresholds (e.g., some studies used p<0.05, while others used p<0.01). Please clarify.
- Results: Table 4. Please add footnotes explaining the abbreviations (PCoA, PERMANOVA, etc.).
- Results: Table 5. Please explain the abbreviations ESCC, EC, OTU, EAC.
- Results: A meta-analysis was performed for two independent studies. Please explain why only two studies were eligible for this analysis.
- Discussion: Please include the CASP quality assessment results in the discussion. In addition, please indicate whether age, gender, smoking, and alcohol consumption were taken into account in these studies.
- Discussion: In the section on alpha diversity, please refer to the previous results. Was alpha diversity influenced by differences in sample types (saliva vs swabs), sequenced regions (V3–V4 vs V4) and platforms (MiSeq vs Ion Torrent)?
- Discussion. Please check and correct the italicised genes: TP53, RB1, CDKN2A, PIK3CA, NOTCH1, and NFE2L2, as well as the names of bacteria, e.g. Streptococcus bovis, Prevotella.
- Complete the abbreviations section.
- From July 2024 to September 2025, new original publications and review articles on this topic appeared. It is worth checking whether they would enrich Your work.
Author Response
Comment 1:
Materials and methods: The end date of the search was set at 6 July 2024, i.e., mid-2024. Please provide information on whether the search was updated close to the date of manuscript submission. Such information would increase the timeliness of the study.
Response 1:
We thank the reviewer for this important comment. The literature search was initially conducted up to 6 July 2024. Prior to submission, we re-checked PubMed, Embase, and Web of Science in July 2025. Four additional studies were identified with consistant and additional informative findings. We have added this information in our Discussion (Page16, Line263)
Comment 2:
Materials and methods: It would be useful to supplement this section with more detailed information on the meta-analysis methods. The information provided is not detailed enough. Please explain why the fixed-effect model was chosen? Was heterogeneity assessed using specific tests?
Response 2:
We thank the reviewer for this important suggestion. In the revised manuscript, we have expanded the description of our meta-analysis methods (Page4, Line164) to clarify the choice of model and the assessment of heterogeneity.
Comment 3:
Materials and methods: Please indicate at this stage how many studies were included in the meta-analysis and whether any were excluded due to insufficient data (in the next section there are 18).
Response 3
Yes, already add the numbers of exclusion and inclusion in Section 2.4. This has been added to avoid confusion and improve transparency.
Comment 4:
Materials and methods: The CASP checklist was used, which is an appropriate choice. However, it would be worth adding information on whether the CASP assessments were carried out independently by two reviewers.
Response 4:
We thank the reviewer for this observation. In the revised manuscript, we now clarify that the CASP quality assessments were carried out independently by two reviewers (J.-C. Chen and M.-H. Hsu). Any discrepancies in scoring were discussed and resolved through consensus with a third reviewer (S.-W. Hu). This information has been added to the Materials and Methods section (Page 5, Line181) for greater transparency and methodological rigor.
Comment 5:
Materials and methods: Please indicate why articles published before 31 December 2010 were excluded (is it about NGS?).
Response 5:
We thank the reviewer for this question. Yes, our rationale was based on methodological considerations. High-throughput next-generation sequencing (NGS) technologies, particularly 16S rRNA gene sequencing and metagenomics, only became widely available and standardized in microbiome research after 2010. Earlier studies primarily relied on culture-based methods or low-throughput molecular techniques, which lack the resolution and comparability required for inclusion in a systematic review focused on oral microbiome profiling. To ensure methodological consistency and comparability across studies, we therefore restricted inclusion to articles published from 2011 onwards. We have clarified this rationale in the Materials and Methods section (Page 4, Line 146).
Comment 6:
Results: Please provide a definition of the exclusion step ‘irrelevant topics’ in the results, in the table - Not related (n = 644); this aspect is unclear. Please consider specifying common topics or categories of irrelevance.
Response 6:
We thank the reviewer for pointing out this ambiguity. In the revised manuscript, we have clarified what was meant by “irrelevant topics.” These were studies that did not directly investigate the oral microbiome in relation to esophageal cancer (Page 5, Line 195). The most common excluded categories included:
- Studies of gastrointestinal microbiota not involving the oral cavity (e.g., gut or gastric microbiome only).
- Studies of oral microbiota unrelated to cancer (e.g., focusing on caries, periodontitis, or oral candidiasis).
- Studies of esophageal cancer not assessing microbial composition (e.g., clinical, genetic, or imaging studies).
- Review articles, editorials, or conference abstracts that did not provide primary microbiome data.
Comment 7:
Results: Table 2 Abbreviations (e.g., ESCC, EAC) should be defined once in a footnote to the table, even if they have been explained earlier in the text.
Response 7:
We thank the reviewer and did so accordingly.
Comment 8:
Results: Table 4. Please try to standardise the entries in the table and remove the inconsistent formatting. Some rows contain detailed statistics (e.g., Shannon values), while others contain only a qualitative description, “No α diff”. Standardisation is recommended (e.g., mean ± SD or at least consistent directional reporting). Abbreviations such as “diff.” or “NS” should be replaced with full terms (“difference”, “not significant”).
Response 8:
We thank the reviewer for the suggestion. The Table 4 has been revised.
Comment 9:
Results: Table 4. The description for Table 4 mentions ‘significant’ results but does not provide consistent p-value thresholds (e.g., some studies used p<0.05, while others used p<0.01). Please clarify.
Response 9:
We thank the reviewer for the suggestion. The Table 4 has been revised.
Comment 10:
Results: Table 4. Please add footnotes explaining the abbreviations (PCoA, PERMANOVA, etc.).
Response 10:
We thank the reviewer for the suggestion. The Table 4 has been revised.
Comment 11:
Results: Table 5. Please explain the abbreviations ESCC, EC, OTU, EAC.
Response 11:
We thank the reviewer for the suggestion. The Table 5 has been revised.
Comment 12:
Results: A meta-analysis was performed for two independent studies. Please explain why only two studies were eligible for this analysis.
Response 12:
We thank the reviewer for raising this important point. Although 18 studies were included in the systematic review, the majority did not provide sufficient or comparable quantitative data for pooling. Specifically, many studies either (i) reported only relative abundance differences without effect size measures or raw numerical data, (ii) used heterogeneous outcome definitions, or (iii) lacked adequate case–control comparability. As a result, only two studies reported effect estimates with appropriate statistical measures that could be synthesized meaningfully in a meta-analysis. We have clarified this in the revised Results and Materials and Methods sections to avoid confusion (Page 12, Line 308).
Comment 13:
Discussion: Please include the CASP quality assessment results in the discussion. In addition, please indicate whether age, gender, smoking, and alcohol consumption were taken into account in these studies.
Response 13:
The following paragraph has been added (Page 16, Line 462).
The CASP quality assessment indicated that most of the included studies were of moderate to high methodological quality, with clearly defined objectives and appropriate designs. However, variability was noted in several domains, particularly group comparability and control for confounders. While many studies accounted for age and gender, fewer consistently adjusted for smoking and alcohol consumption, despite their strong associations with both esophageal cancer risk and oral microbiome composition. In some cases, incomplete reporting of these variables limited the ability to assess their impact. This heterogeneity in adjustment for key confounders represents a significant limitation of the current evidence base, as residual confounding may partly explain the observed microbial associations. Future studies should therefore adopt more rigorous designs with comprehensive recording and adjustment for demographic and lifestyle factors in order to strengthen causal inference.
Comment 14:
Discussion: In the section on alpha diversity, please refer to the previous results. Was alpha diversity influenced by differences in sample types (saliva vs swabs), sequenced regions (V3–V4 vs V4) and platforms (MiSeq vs Ion Torrent)?
Response 14:
We thank the reviewer for the suggestion. We have incorporated alpha diversity in the Discussion (Page 14, Line376).
Comment 15:
Discussion. Please check and correct the italicised genes: TP53, RB1, CDKN2A, PIK3CA, NOTCH1, and NFE2L2, as well as the names of bacteria, e.g. Streptococcus bovis, Prevotella.
Response 15:
We thank the reviewer for the suggestion.
Comment 16:
Complete the abbreviations section.
Response 16:
We thank the reviewer for the suggestion.
Comment 17:
From July 2024 to September 2025, new original publications and review articles on this topic appeared. It is worth checking whether they would enrich Your work.
Response 17:
We thank the reviewer for this helpful suggestion. We reviewed the recent publications from July 2024 to September 2025 and agree that they provide important insights that enrich our manuscript. Accordingly, we have incorporated the following updates (Page 16, Line 474 and Page 17, Line 511):
Our initial search covered studies published up to mid 2024. The databases were re-checked in July 2025 prior to submission; 4 additional studies were identified with consistant and additional informative findings [74–77]. The oral microbiome also shows promise as a non-invasive biomarker for EC. Predictive models based on microbial profiles have achieved high accuracy, with an ESCC classifier reaching an AUC of 0.87. Key microbial markers include Neisseria perflava and Haemophilus parainfluenzae. Saliva-based prediction models also demonstrated strong discriminative ability (AUC = 0.791), with decreased abundance of Akkermansia and Escherichia-Shigella linked to increased cancer risk. These findings underscore the translational potential of microbial markers for early detection.
We thank the reviewer again for directing us to these valuable new resources, which have strengthened and updated our manuscript.
Round 2
Reviewer 1 Report
Comments and Suggestions for Authors
Authors have addressed all of my concerns, it could be accepted for publication now.
Reviewer 2 Report
Comments and Suggestions for Authors
Thank You for Your responses to my comments. Based on Your answers and the changes made to the publication, I believe that the publication can be accepted.
I wish You continued success in the future.
line 170 - correct metaq-analysis